# Understanding Before Evaluation: A Reliable Framework for Assessing Non-Deterministic Machine Translation Systems

## Abstract

Modern machine translation (MT) systems exhibit non-deterministic behavior, producing variant outputs across runs in both neural MT and LLM-based MT. This variability poses significant challenges for automatic evaluation methods (AEMs), leading to unreliable quality assessments. To address this limitation, we propose a two-stage **"Understanding Before Evaluation"** framework. In the understanding stage, we formalize and measure the degree of non-determinism from both lexical and semantic perspectives using a simple sample-based strategy. Comprehensive experiments on public datasets reveal high variance in lexical-based methods while demonstrating stable behavior in semantic-based approaches across MT systems. In the evaluation stage, we propose a reliable *ExpectoSample* method that explicitly incorporates non-deterministic characteristics to mitigate variance effects. Our two-stage framework delivers more reliable quality assessments for modern MT systems. Furthermore, our methods provide a potential way for measuring MT metrics without human involvement and highlight the superiority of semantic-based metrics for evaluating modern non-deterministic MT systems.

## 1 Introduction

Machine translation (MT) has advanced rapidly in recent decades, driven by neural MT (Devlin et al., 2019; Lewis et al., 2020; Raffel et al., 2020; Costa-jussà et al., 2022) and large language models (LLMs) (Chowdhery et al., 2023; OpenAI, 2023; Touvron et al., 2023; Yang et al., 2024; DeepSeek-AI et al., 2025; Yang et al., 2025; Vilar et al., 2023; Bawden & Yvon, 2023; Moslem et al., 2023). However, the reliable evaluation of MT systems remains challenging (Kocmi et al., 2024). Current evaluation typically applies test sets and automatic metrics (Kocmi et al., 2025), testing the correlation with human judgments (Kocmi et al., 2024). Unfortunately, such a human-involved scheme still fails for MT systems when facing non-determinism (Reimers & Gurevych, 2018; Sanchez Carmona et al., 2025).

Modern MT relies on probability-based generation with an attention mechanism Vaswani et al. (2017), where outputs are sampled from a softmax distribution. This introduces a natural non-determinism that will finally propagate to sequence-level candidates. Prior studies (Leblond et al., 2021; He & Lab, 2025; Atil et al., 2024) show that non-determinism is intrinsic to MT systems. Evaluating only a single sampled candidate, as in the common generate-once scheme, will risk biased assessment by ignoring equally generated non-deterministic candidates and lead to unreliable evaluation results.

Measuring non-determinism directly is difficult. Token-level alignments are unobservable because many semantic alignments exist beyond the token-level. Moreover, multiple valid references exist (Papineni et al., 2002; Freitag et al., 2020), making the counting of correct references impossible. Additionally, the unpredictable sequence length makes it hard even for a theoretical analysis. Entropy-based approaches (Guerreiro et al., 2023; Yeom et al., 2018; Carlini et al., 2021; Shi et al., 2024; Zhang et al., 2024) try to understand the non-deterministic behavior of LLMs but assume strong memorization (Shi et al., 2024) without

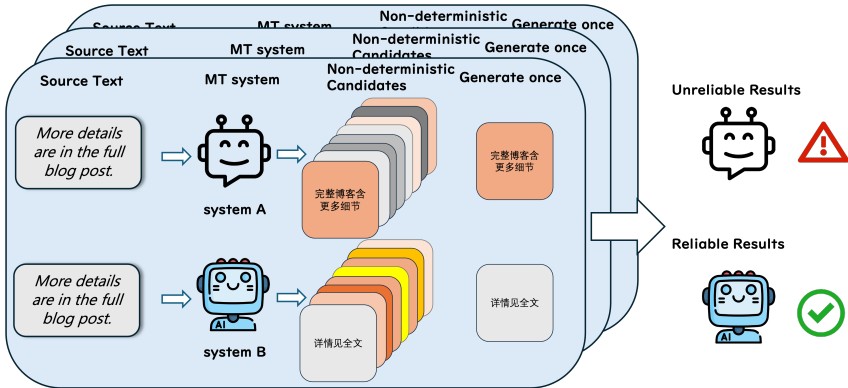

Figure 1: Illustration of non-deterministic MT systems. For one source, multiple candidates can be generated. Grey blocks indicate low-quality outputs; lighter blocks indicate high-quality ones. Generate-once evaluation may misrepresent system quality if only some candidates are sampled.

considering semantic alignment that is crucial for MT systems. Semantic uncertainty (Kuhn et al., 2023b; Qiu & Miikkulainen, 2024; Jia et al., 2025) uncovers the semantic similarity within candidates while ignoring the alignment with the input. To overcome the gap in the understanding and measuring non-determinism of MT systems, we propose non-determinism degree measurements from lexical and semantic perspectives with a simple sampling strategy. This approach yields interpretable linguistic insights and quantitative scores with either candidates themselves or existing metrics.

Building on this, we introduce the **Understanding Before Evaluation (UBE)** framework, which incorporates non-determinism into MT evaluation. Our systematic experiments on SOTA MT systems show the effectiveness of our designed understanding measurements. We further design a method named *ExpectoSample* through utilizing a sampling-based expectation strategy that keeps the evaluation reliability across multiple metrics and sampling sizes. Our findings highlight non-determinism as inherent and unavoidable, and call for developing semantic-oriented metrics beyond costly human evaluation (Graham et al., 2013; Lommel et al., 2014). Crucially, our strategy is lightweight and compatible with current APIs and open-source MT systems. Our contributions are:

- We demonstrate that non-determinism is intrinsic to modern MT systems and undermines existing evaluation practices.

- We propose measurements of non-determinism on both lexical and semantic perspectives through an efficient sampling strategy.

- We introduce the **Understanding Before Evaluation (UBE)** frame to bridge the ignorance of the deterministic nature of MT systems and propose the *ExpectoSample* strategy for reliable MT system evaluation.

- Our frame and strategy are valuable for validating the reliability of MT metrics. Besides, our findings unveil the trend and value of developing semantic-based metrics for MT.

In summary, we unveil the truth that non-deterministic MT systems have the ability to generate candidates with flexible lexical and well-studied semantic alignment behavior =. (Chu & Wang, 2018), which risks the reliable evaluation of MT systems, and we can solve it through the measurement of non-deterministic degree. We call for considering non-deterministic degree measurement for all current developed MT systems and re-evaluate the ranking of MT systems beyond the alignment design with human assessment (Graham et al., 2013; Lommel et al., 2014).

## 2 THE FORMALIZATION OF THE NON-DETERMINISTIC DEGREE OF MT SYSTEMS

### 2.1 CHALLENGES OF IGNORING NON-DETERMINISM IN EVALUATION

As shown in Figure 1, overlooking the non-deterministic nature of MT systems leads to unreliable evaluations across systems. At the instance level, it may also mislead users to underestimate the ability of an MT system to produce lexically rich candidates. In this section, we first explain why non-determinism cannot be directly captured from token-level probabilities, and then introduce measurements of non-deterministic degree on both lexical and semantic aspects that enable both qualitative interpretation and quantitative assessment.

### 2.2 CHALLENGES IN UNDERSTANDING AND MEASURING NON-DETERMINISM

One intuitive approach for quantifying non-determinism involves leveraging token probabilities (Kuhn et al., 2023b). Nevertheless, this methodology proves impractical for MT since MT is a sequential generation task characterized by variable output lengths (Freitag et al., 2020). Furthermore, establishing semantic correspondence between individual tokens and source content presents significant challenges (Kudo & Richardson, 2018).

Previous research (Atil et al., 2024) has investigated question-answering tasks (Joshi et al., 2017; Hendrycks et al., 2021), where exact answer matching provides a viable evaluation framework, but such approaches cannot be readily adapted to machine translation scenarios with no clear answers. Alternative methodologies estimate entropy across model outputs (Guerreiro et al., 2023; Yeom et al., 2018; Carlini et al., 2021; Shi et al., 2024; Zhang et al., 2024), operating under the assumption that entropy can express the faithfulness of LLMs but without considering semantic alignments between candidates and source texts. A parallel research direction investigates semantic uncertainty (Kuhn et al., 2023b; Qiu & Miikkulainen, 2024; Jia et al., 2025), employing established similarity models (Conneau et al., 2018a) to quantify internal semantic coherence. However, these approaches fail to account for source-target alignment and present substantial challenges when extended to sequential MT lacking explicit lexical correspondences.

### 2.3 NON-DETERMINISTIC DEGREE OF MT SYSTEMS FROM LEXICAL AND SEMANTIC PERSPECTIVES

Understanding the meaning of such a non-deterministic degree (Reimers & Gurevych, 2018; Sanchez Carmona et al., 2025) and the interpretation (Kocmi et al., 2024) of the final score are both significant for MT systems. We consider two main aspects: lexical and semantic. On the lexical, we can check with humans and encompass the linguistic knowledge (Nguyen & Chiang, 2018). On the semantic level, we follow the requirement of semantic alignment of MT with external tools, testing the non-deterministic degree on the semantic view (Rei et al., 2020; 2022; Heffernan et al., 2022; Song et al., 2020; Conneau et al., 2020; Dale & Costa-jussà, 2024; Duquenne et al., 2023).

We now introduce the basic formalism for measuring non-determinism. For an MT task $\mathcal{T}$, let the test set be $\mathcal{S}$ of size $k$, where each source text is $s \in \mathcal{S}$. A non-deterministic MT system $m_\theta \in \mathcal{M}_\Theta$ with parameters $\theta$ can generate, for a source $s_i$, a candidate set $C_{s_i} = \{c_1, c_2, \ldots, c_N\}$. Ideally, there exists a set of gold references $R_{s_i} = \{r_1, r_2, \ldots, r_M\}$. We denote the non-determinism degree for a single source as $d(s_i)$, and for the entire test set:

$$D(\mathcal{S}) = \frac{1}{k} \sum_{s_i \in \mathcal{S}} d(s_i). \tag{1}$$

In practice, neither the full set of candidates nor exhaustive gold references are obtainable, even with human annotation. Following previous work on sampling strategies (Kuhn et al., 2023b) and empirical observations of MT generation (Kocmi et al., 2024), we approximate

$C_{s_i}$ by sampling $n$ candidates under a decoding configuration $\theta_p$, and assume one reference $r_i$ per source $s_i$. We then evaluate $d(s_i)$ from both lexical and semantic perspectives.

**Lexical Non-determinism Degree of MT Systems**  Previous approaches (Papineni et al., 2002) typically employ an anchor, such as a reference translation, to evaluate the lexical quality of generated candidates. However, this method can be problematic for measuring non-determinism when the reference contains highly idiosyncratic lexical choices, leading to little or no lexical overlap. To mitigate this limitation and reduce potential reference bias, we propose an intrinsic method, denoted as **INNER**, which evaluates the degree of lexical non-determinism solely based on the generated candidates themselves. The procedure consists of four steps:

**Step1**: tokenizing each candidate $c_i$ into a list of words $\mathcal{W}_i = \{w_1, w_2, \ldots, w_l\}$

**Step 2**: Construct the frequency vocabulary $\mathcal{V}_i$ from the set of candidates.

**Step 3**: Compute the inner-overlap score $L(c_i)$ for each candidate $c_i$ based on $\mathcal{V}_i$:

$$L(c_i) = \sum_{w \in \mathcal{W}_i^U} f_{\mathcal{V}_i}(w), \tag{2}$$

where $\mathcal{W}_i^U$ denotes the set of unique words in $c_i$, and $f_{\mathcal{V}_i}(w)$ is the frequency of word $w$ in the overall vocabulary $\mathcal{V}_i$.

**Step 4**: Aggregate the scores across sampled candidates $\{L(c_1), L(c_2), \ldots, L(c_n)\}$ to estimate the degree of lexical non-determinism:

$$d(s_i, m_{\theta_p}) = \text{mean\_std}\big(\{L(c_1), L(c_2), \ldots, L(c_n)\}\big),$$

where mean_std computes both the mean and standard deviation of the score distribution. The resulting pair $(\text{mean}, \text{std})$ reflects the degree of non-determinism for a given source. For the full test set, we define:

$$D(\mathcal{S}) = \frac{1}{k} \sum_{s_i \in \mathcal{S}} d(s_i, m_{\theta_p}). \tag{3}$$

However, in most cases, the reference tends to show at least some of the overlap since some unique lexical items can determine the meaning in languages. Following this convention, we consider an external measure of the lexical degree in a relative way with current lexical similarity metrics $\textbf{Sim}_{\textbf{lex}}(\cdot)$

$$d(s_i, m_{\theta_p}, r_i, \textbf{Sim}_{\textbf{lex}}(\cdot)) = \text{mean\_std}(\textbf{Sim}_{\textbf{lex}}(c_i, r_1), \ldots, \textbf{Sim}_{\textbf{lex}}(c_i, r_n)) \tag{4}$$

**Semantic non-determinism degree of MT systems**  we follow the fundamental requirement of MT on the semantic alignment. We utilize the external tools (Rei et al., 2020; 2022; Heffernan et al., 2022; Song et al., 2020; Conneau et al., 2020; Dale & Costa-jussà, 2024; Duquenne et al., 2023) to measure the semantic alignment on the candidate list $C_i$ since the semantics cannot be directly understood with a pure lexical expression. Here we consider both reference-based:

$$d(s_i, m_{\theta_p}, r_i, \textbf{Sim}_{\textbf{sem}}(\cdot) = \text{mean\_std}(\textbf{Sim}_{\textbf{sem}}(s_i, c_i, r_1), \ldots, \textbf{Sim}_{\textbf{sem}}(s_i, c_i, r_n)) \tag{5}$$

and reference free format:

$$d(s_i, m_{\theta_p}, r_i, \textbf{Sim}_{\textbf{sem}}(s_i, c_i) = \text{mean\_std}(\textbf{Sim}_{\textbf{sem}}(s_i, c_1), \ldots, \textbf{Sim}_{\textbf{sem}}(s_i, c_n)) \tag{6}$$

For the measurement of the non-determinism degree of the MT system, we compute with the equation 3.

## 3 Understanding Before Evaluation

Ignoring the non-deterministic nature of MT systems poses the risk of unreliable evaluation. To address this challenge, we propose the **Understanding Before Evaluation (UBE)** framework, which separates MT evaluation into two stages:

**Understanding Stage** In this stage, we employ the formal definitions introduced in Section 2 to calculate the non-deterministic degree of an MT system from both lexical and semantic perspectives. We further analyze these degrees for each system and treat the resulting scores as intrinsic attributes of the systems under investigation.

**Evaluation Stage** To mitigate the unreliability caused by ignoring non-determinism, we introduce the **ExpectoSample** strategy. This method can be directly applied to existing MT metrics, enabling more reliable evaluations of non-deterministic systems.

### 3.1 EXPERIMENTAL SETUP

**Data:** We adopt the WMT23 (Kocmi et al., 2023) and WMT24 (Wang et al., 2024a) datasets[12], focusing on the Chinese–English language pair, which comprises 5,048 sentence pairs.

**Models and Prompting:** For a comprehensive evaluation, we include representative state-of-the-art transformer-based (Vaswani et al., 2017) non-deterministic MT systems. These cover (i) conventional NMT systems (Lewis et al., 2020; Costa-jussà et al., 2022), (ii) pre-trained LLMs (Touvron et al., 2023; Yang et al., 2024), (iii) instruction-tuned LLMs (Touvron et al., 2023; Yang et al., 2024; Hu et al., 2024), and (iv) reasoning-focused LLMs (DeepSeek-AI et al., 2025; Yang et al., 2025). We detail the information of all model in Appendix A

For prompting, we adopt Five-Shot prompting to reduce potential mismatches and repetitive patterns (Wang et al., 2024b). Instruction-tuned LLMs are evaluated with direct prompts in order to avoid artificial biases while leveraging their intrinsic translation capability (Vilar et al., 2023). Similarly, reasoning LLMs are prompted directly, ensuring their reasoning is focused solely on the MT task. Detailed model configurations and prompt designs are provided in the Appendix B.

**Sampling Strategy:** Motivated by prior findings on temperature' s effect on generation (Kuhn et al., 2023a), we set the temperature to 0.5 and sample 10 candidates per source. Importantly, our method is not constrained to this specific setting and can be flexibly adapted to other generation configurations.

**Non-determinism Degree Measurement:** To compute the non-deterministic degree, especially from the semantic perspective, we rely on external evaluation tools. For lexical measurements, we adopt BLEU (Papineni et al., 2002), METEOR (Banerjee & Lavie, 2005), chrF++ (Popović, 2017), TER (Snover et al., 2006), ROUGE (Lin, 2004), BERTScore (Zhang et al., 2019), and BLEURT (Sellam et al., 2020). For semantics, we consider supervised MT metrics such as COMET20DA (Rei et al., 2020) and COMET22KIWI (Rei et al., 2022); supervised sentence-similarity models like LaBSE (Feng et al., 2020) and XNLI (Conneau et al., 2018b); as well as embedding-based unsupervised methods including LASER (Heffernan et al., 2022), Sentence Transformers (Song et al., 2020), SONAR (Duquenne et al., 2023), and BLASER (Dale & Costa-jussà, 2024).

**MT Metrics:** For final MT evaluation, we directly employ the metrics computed during non-determinism measurement, excluding those pure similarity measures that are not supervised with MT data (i.e., LaBSE (Feng et al., 2020), XNLI (Conneau et al., 2018b), and the embedding-based unsupervised methods LASER (Heffernan et al., 2022), Sentence Transformers (Song et al., 2020), SONAR (Duquenne et al., 2023), and BLASER (Dale & Costa-jussà, 2024)).

Table 1: The results of non-determinism degrees show high lexical variance and stable semantic alignment.

| Model Name | Degree of Non-determinism | | | | | |
| | INNER | | COMET20DA | | COMET22KIWI | |
| | mean | std | mean | std | mean | std |
|---|---|---|---|---|---|---|
| LlaMA-2-7b-Instruct | 58.35 | 5.19 | 66.61 | 7.89 | 58.51 | 10.00 |
| LlaMA-2-7b-Pre | 46.23 | 13.39 | 70.67 | 8.00 | 63.80 | 8.31 |
| Qwen2.5-7b-Instruct | 85.65 | 10.00 | 85.15 | 2.06 | 79.01 | 2.10 |
| Qwen2.5-7b-Pre | 56.89 | 19.64 | 77.45 | 7.60 | 75.09 | 5.85 |
| Qwen3-8b-NT | 90.66 | 8.05 | 86.24 | 1.20 | 80.68 | 0.99 |

## 3.2 UNDERSTANDING STAGE

### 3.2.1 CASE STUDY: ANALYSIS OF LEXICAL AND SEMANTIC DEGREES OF NON-DETERMINISM

In the understanding stage, we aim to characterize the non-deterministic behavior of MT systems in a measurable way. For this case study, we consider three representative systems: LlaMA2 (Touvron et al., 2023), Qwen2.5 (Yang et al., 2024), and Qwen3 (Yang et al., 2025). Table 1 reports their non-determinism measured from both the lexical perspective (**INNER**, defined in Section 2) and the semantic perspective, using two external MT metrics (COMET20DA and COMET22KIWI). Each system is evaluated by the mean and standard deviation across sampled candidates.

From the lexical perspective, the **mean** of INNER reflects the degree of lexical flexibility, while the **standard deviation (STD)** captures variability or extreme cases. For example, Qwen2.5-7b-Pre shows a relatively low mean (56.89) but the largest STD (19.64), indicating that while it sometimes generates diverse lexical alternatives, its outputs are highly unstable. Similarly, LlaMA-2-7b-Pre reaches an even lower mean (46.23) yet still a very high STD (13.39), pointing to unstable lexical diversity and occasional extreme outputs such as malformed translations. Such behavior warrants careful attention in applications where stability of surface forms is critical.

From the semantic perspective, the distributions are more robust, especially with respect to STD values. Here, the **mean** serves as an indicator of semantic adequacy, where higher values imply stronger alignment between source and candidate. Ideally, candidates should reach both high means and low STDs, reflecting semantically faithful and consistent outputs. Qwen3-8B-NT exemplifies this behavior: it achieves the highest COMET20DA score (86.24, STD 1.20) and the highest COMET22KIWI score (80.68, STD 0.99), demonstrating strong and stable semantic alignment. Nevertheless, it should be noted that these semantic scores are derived from external neural metrics, which may introduce their own biases, as highlighted in recent work (Zouhar et al., 2024).

Overall, this case study illustrates system-level trade-offs. For applications prioritizing semantic quality and stability, Qwen3-8B-NT (Yang et al., 2025) is the preferred choice, given its consistently high adequacy and low variance. When applications benefit from both high quality and greater lexical flexibility, Qwen2.5-7b-Pre (Yang et al., 2024) offers a more balanced option, albeit with higher variability. By contrast, the LlaMA-2-7b systems show weaker overall performance, with comparatively lower means and higher variability in both lexical and semantic dimensions.

## 3.3 THE UNRELIABILITY OF GENERATE-ONCE EVALUATION WITH NON-DETERMINISTIC MT SYSTEMS

An important implication of measuring non-determinism is that it highlights the inherent unreliability of current *generate-once* evaluation, even when advanced metrics are ap-

---

[1]https://github.com/wmt-conference/wmt23-news-systems
[2]https://github.com/wmt-conference/wmt23-news-systems

plied (Freitag et al., 2024). As shown in Table 2, relying on a single sampled output can shift system rankings. Furthermore, the divergence between rankings based on mean scores and those from arbitrary random samples indicates that repeated generation alone does not guarantee robust comparisons across non-deterministic MT systems.

Table 2: The systematic results of the degree of non-determinism show the risk of unreliable ranking.

| Model Name | Size | BLEU | | | | TER | | | | COMET20DA | | | | COMET22KIWI | | | |
|---|---|---|---|---|---|---|---|---|---|---|---|---|---|---|---|---|---|
| | | max | mean | min | rand | max | mean | min | rand | max | mean | min | rand | max | mean | min | rand |
| NMT | | | | | | | | | | | | | | | | | |
| MBART | 610M | 13 | 12 | 13 | 14 | 13 | 15 | 13 | 14 | 13 | 11 | 10 | 11 | 14 | 13 | 11 | 13 |
| NLLB-200 | 600M | 20 | 19 | 16 | 19 | 16 | 16 | 7 | 15 | 18 | 18 | 15 | 18 | 17 | 16 | 16 | 16 |
| NLLB-200 | 3.3B | 17 | 16 | 17 | 17 | 14 | 10 | 5 | 10 | 15 | 14 | 13 | 14 | 15 | 15 | 15 | 15 |
| NLLB-moe | 54.5B | 19 | 18 | 18 | 18 | 15 | 14 | 6 | 12 | 16 | 15 | 16 | 15 | 16 | 18 | 18 | 18 |
| LLM (pre-trained only) | | | | | | | | | | | | | | | | | |
| Llama-2 | 7B | 8 | 13 | 12 | 13 | 10 | 13 | 16 | 13 | 11 | 13 | 14 | 13 | 13 | 14 | 13 | 14 |
| Qwen2.5 | 7B | 16 | 20 | 20 | 20 | 1 | 1 | 4 | 1 | 5 | 9 | 11 | 10 | 3 | 7 | 8 | 7 |
| Llama-3.1 | 8B | 12 | 15 | 19 | 16 | 3 | 7 | 17 | 7 | 10 | 10 | 12 | 9 | 12 | 11 | 12 | 11 |
| Llama-2 | 70B | 11 | 14 | 14 | 12 | 7 | 6 | 15 | 6 | 17 | 16 | 17 | 16 | 19 | 17 | 17 | 17 |
| Llama-3.1 | 70B | 9 | 5 | 6 | 6 | 4 | 3 | 1 | 3 | 14 | 12 | 9 | 12 | 10 | 8 | 7 | 8 |
| Qwen2.5 | 72B | 18 | 17 | 15 | 15 | 2 | 2 | 14 | 2 | 12 | 17 | 19 | 17 | 11 | 12 | 14 | 12 |
| LLM (instruction-tuned) | | | | | | | | | | | | | | | | | |
| Llama-2 | 7B | 14 | 10 | 8 | 10 | 6 | 5 | 2 | 5 | 19 | 20 | 20 | 20 | 18 | 19 | 20 | 19 |
| Qwen2.5 | 7B | 6 | 4 | 3 | 3 | 17 | 17 | 9 | 16 | 3 | 1 | 1 | 1 | 4 | 1 | 1 | 1 |
| Llama-2 | 70B | 10 | 11 | 10 | 11 | 5 | 4 | 3 | 4 | 20 | 19 | 18 | 19 | 20 | 20 | 19 | 20 |
| Qwen2.5 | 72B | 4 | 1 | 1 | 1 | 20 | 20 | 11 | 18 | 6 | 4 | 4 | 4 | 6 | 4 | 4 | 4 |
| MiniCPM-MoE | 8x2B | 15 | 9 | 5 | 9 | 18 | 18 | 12 | 19 | 8 | 6 | 5 | 6 | 7 | 6 | 5 | 5 |
| LLM (reasoning) | | | | | | | | | | | | | | | | | |
| Qwen3(NT) | 8B | 7 | 2 | 2 | 2 | 19 | 19 | 8 | 20 | 4 | 2 | 2 | 2 | 5 | 2 | 2 | 2 |
| Qwen3 | 8B | 2 | 3 | 4 | 4 | 11 | 12 | 18 | 17 | 2 | 3 | 3 | 3 | 1 | 3 | 3 | 3 |
| DeepSeek-R1 | 7B | 1 | 7 | 9 | 8 | 8 | 8 | 10 | 8 | 9 | 8 | 8 | 8 | 9 | 10 | 10 | 10 |
| DeepSeek-R1 | 8B | 3 | 8 | 11 | 5 | 9 | 9 | 19 | 9 | 7 | 7 | 7 | 7 | 8 | 9 | 9 | 9 |
| DeepSeek-R1 | 671B | 5 | 6 | 7 | 7 | 12 | 11 | 20 | 11 | 1 | 5 | 6 | 5 | 2 | 5 | 6 | 6 |

## 3.4 EVALUATION STAGE

The degree of non-determinism observed in the *Understanding Stage* makes it impractical to rely on a generate-once for system evaluation. The MT system can output multiple candidates due to its non-deterministic nature. To solve the unreliable situation, we propose an Expectation to the Samples (*ExpectoSample*) strategy that can incorporate any MT metrics to consider the non-determinism and provide a reliable evaluation.

Formally, for an MT metric $M(\cdot)$ applied to translated outputs, given a set of $n$ sampled candidates $\{y_1, y_2, \ldots, y_n\}$, the evaluation score for a source text $s_i$ is estimated as:

$$Eval(s_i) = \mathbb{E}[Met\,(s_i, y_j, r_i)] = \frac{1}{n} \sum_{j=1}^{n} Met\,(s_i, y_j, r_i). \tag{7}$$

For the whole evaluation set $\mathcal{S}$ with size $k$, the final score for the MT system $m_{\theta_p}$ is:

$$Eval(S) = \mathbb{E}[Eval(s_i)] = \frac{1}{k} \sum_{i=1}^{k} Eval(s_i) \tag{8}$$

This expectation over samples directly reduces estimator bias, a standard technique to mitigate randomness for non-deterministic systems. Intuitively, the ExpectoSample allows the flexibility of lexical and measures the semantic alignment on both the instance-level (with E.q. 7) and system-level (with E.q. 8). Besides, the sampling strategy is easy to implement for any non-deterministic MT system without much additional cost for a small sampling size.

## 3.5 THE RELIABILITY OF EXPECTOSAMPLE STRATEGY

One key factor for our method is the choice of sampling size $n$. Larger values of $n$ will affect the non-determinism degree and may lead to a change in the ranking. We select three sample sizes ($n = \{10, 20, 50\}$ to validate the reliability of our proposed method.

We keep the rest setting used in the measurement of the non-determinism degree and choose LlaMA2-7b (Touvron et al., 2023), Qwen2.5-7b (Yang et al., 2024) and Qwen3-8b (Yang et al., 2025) models under EN-ZH settings on WMT23 data to test the change. From the above case study and the ranking results in Table 1, and 2, we already observe the huge difference for these five systems, so we select them as the standard models to do such exploration.

Table 3 shows that our proposed method yields consistent performance rankings across multiple MT metrics. In particular, we observe that both lexical and semantic metrics produce identical ranking results under different sample sizes. These findings suggest that ExpectoSample is reliable for evaluating diverse machine translation systems. At the same time, Table 4 reveals some irregular patterns in metrics such as TER and ROUGE, indicating that our ExpectoSample strategy can also serve to assess the reliability of existing machine translation metrics. ExpectoSample is computationally inexpensive in additional sampling and can seamlessly integrate with any MT metrics. Furthermore, our method is unsupervised without the need for human assessment to evaluate MT systems.

Table 3: Evaluation on the effect of sample size with consistent rankings.

| | Model Name | BLEU | BERTScore | chrF++ | COMET20DA | COMET22KIWI | BLEURT |
|---|---|---|---|---|---|---|---|
| **Sampling Size=10** | Llama-2-7b-Instruct | 3 | 5 | 5 | 5 | 5 | 5 |
| | Llama-2-7b-Pre | 4 | 4 | 4 | 4 | 4 | 4 |
| | Qwen2.5-7b-Instruct | 1 | 2 | 2 | 2 | 2 | 2 |
| | Qwen2.5-7b-Pre | 5 | 3 | 3 | 3 | 3 | 3 |
| | Qwen3-8b-NT | 2 | 1 | 1 | 1 | 1 | 1 |
| **Sampling Size=20** | Llama-2-7b-Instructt | 3 | 5 | 5 | 5 | 5 | 5 |
| | Llama-2-7b-Pre | 4 | 4 | 4 | 4 | 4 | 4 |
| | Qwen2.5-7b-Instruct | 1 | 2 | 2 | 2 | 2 | 2 |
| | Qwen2.5-7b-Pre | 5 | 3 | 3 | 3 | 3 | 3 |
| | Qwen3-8b-NT | 2 | 1 | 1 | 1 | 1 | 1 |
| **Sampling Size=50** | Llama-2-7b-Instruct | 3 | 5 | 5 | 5 | 5 | 5 |
| | Llama-2-7b | 4 | 4 | 4 | 4 | 4 | 4 |
| | Qwen2.5-7B-Instruct | 1 | 2 | 2 | 2 | 2 | 2 |
| | Qwen2.5-7b | 5 | 3 | 3 | 3 | 3 | 3 |
| | Qwen3-8B(No Thinking) | 2 | 1 | 1 | 1 | 1 | 1 |

Table 4: Evaluation on the effect of sample size with inconsistent rankings.

| | Model Name | METEOR | ROUGE-1 | ROUGE-L | TER |
|---|---|---|---|---|---|
| **Sampling Size=10** | Llama-2-7b-Instruct | 5 | 4 | 4 | 2 |
| | Llama-2-7b | 4 | 5 | 5 | 3 |
| | Qwen2.5-7B-Instruct | 2 | 2 | 2 | 4 |
| | Qwen2.5-7b | 3 | 3 | 3 | 1 |
| | Qwen3-8B(No Thinking) | 1 | 1 | 1 | 5 |
| **Sampling Size=20** | Llama-2-7b-Instruct | 5 | 4 | 4 | 1 |
| | Llama-2-7b | 4 | 5 | 5 | 2 |
| | Qwen2.5-7B-Instruct | 1 | 1 | 1 | 2 |
| | Qwen2.5-7b | 3 | 3 | 3 | 1 |
| | Qwen3-8B(No Thinking) | 2 | 2 | 2 | 1 |
| **Sampling Size=50** | Llama-2-7b-Instruct | 5 | 5 | 5 | 1 |
| | Llama-2-7b | 4 | 4 | 4 | 2 |
| | Qwen2.5-7B-Instruct | 2 | 2 | 2 | 2 |
| | Qwen2.5-7b | 3 | 3 | 3 | 1 |
| | Qwen3-8B(No Thinking) | 1 | 1 | 1 | 1 |

## 4  RELATED WORKS

**Non-deterministic MT systems**   Non-deterministic MT systems may produce different outputs for the same input when sampling-based decoding is used. Most modern systems fall into this category, as their predictions are probability-based over a fixed vocabulary with attention mechanisms (Vaswani et al., 2017). Current MT systems can be broadly divided into neural MT (NMT) (Dabre et al., 2020) and LLM-based MT (Vilar et al., 2023). NMT typically employs encoder–decoder architectures (Raffel et al., 2020) to learn semantic alignment from training data. For example, NLLB-200 (Costa-jussà et al., 2022) supports over 200 languages using large-scale bitext resources and back-translation.

In contrast, LLM-based MT has recently emerged as a promising paradigm. Pre-trained LLMs often act as strong translators with few-shot prompting (Vilar et al., 2023; Bawden & Yvon, 2023), and their translation capability further improves with instruction tuning or reinforcement learning. In this paper, we consider mainstream families such as LLaMA (Touvron et al., 2023; Dubey et al., 2024) and Qwen (Yang et al., 2024; 2025), covering pre-trained, instruction-tuned, and reasoning-enhanced variants. All these exhibit non-deterministic behavior and thus are suitable for our study.

**Measuring non-determinism in MT**  Non-determinism has been observed even under nominally deterministic settings (Sanchez Carmona et al., 2025), and recent work attributes this to underlying kernels that may be stabilized (He & Lab, 2025). Unlike tasks with unique outputs, such as QA (Joshi et al., 2017) or reasoning benchmarks (Hendrycks et al., 2021), MT permits multiple semantically valid translations. Hence, strict matching criteria used in earlier studies are unsuitable for MT. One line of research measures output uncertainty via entropy (Guerreiro et al., 2023; Yeom et al., 2018; Carlini et al., 2021; Shi et al., 2024; Zhang et al., 2024), assuming entropy faithfully reflects generation confidence. However, this assumption is hard to validate for MT, and entropy-based results are often opaque to human interpretation. Another line focuses on semantic uncertainty (Kuhn et al., 2023b; Qiu & Miikkulainen, 2024; Jia et al., 2025), measuring similarity among generated samples. Yet two limitations remain: (1) gold references are difficult to establish, as reference-based evaluation is biased (Kocmi et al., 2024); and (2) such methods neglect the source sentence, thus overlooking semantic alignment. To address these gaps, we propose measuring non-determinism from both lexical and semantic perspectives.

**Automatic MT evaluation methods**  Automatic evaluation is essential for assessing translation quality and guiding MT development. Existing methods largely fall into lexical- and semantic-based categories. Lexical metrics such as BLEU (Papineni et al., 2002), METEOR (Banerjee & Lavie, 2005), chrF++ (Popović, 2017), TER (Snover et al., 2006), and ROUGE (Lin, 2004) quantify n-gram or character-level overlap. Embedding-based metrics such as BERTScore (Zhang et al., 2019) and BLEURT (Sellam et al., 2020) compute similarity using contextual representations from pre-trained models (Devlin et al., 2019). For semantic alignment, supervised approaches like COMET20DA (Rei et al., 2020) and COMET22KIWI (Rei et al., 2022) are widely used, while XCOMET (Guerreiro et al., 2024) integrates the MQM (Lommel et al., 2014) scheme for fine-grained evaluation. The above methods are typically single-generation-based methods and ignore the non-determinism of MT systems.

## 5 CONCLUSION

In this work, we demonstrate that the non-deterministic nature of MT systems leads to unreliable evaluation across different MT systems under the generate-once strategy. To address this risk, we first define the degree of determinism from both lexical and semantic perspectives for quality analysis and quantitative usage, such as system ranking. Subsequently, we propose an easy-to-implement strategy named ExpectoSample that computes the expectation of candidates sampled according to source texts to mitigate the effects of the non-determinism degree. Our experiments demonstrate that this strategy proves reliable across different sample size settings and can serve as an unsupervised method to assess the reliability of MT metrics without human involvement. Furthermore, our experiments also reveal the robustness of semantic-based MT metrics and highlight the strong capability of non-deterministic MT systems in semantic alignment.

## 6 STATEMENT

### 6.1 ETHICS STATEMENT

Our work focuses on improving the reliability of machine translation (MT) evaluation by explicitly incorporating the non-deterministic nature of modern MT systems. While our framework and ExpectoSample strategy aim to provide trustworthy assessments, we acknowledge that any evaluation framework may still introduce biases depending on the choice of datasets, metrics, or sampling configurations. In particular, incorrect interpretation of non-determinism measurements could mislead practitioners about the actual reliability of MT systems. We emphasize that our contributions are intended for research and evaluation purposes, not as a replacement for human judgment in high-stakes or sensitive domains. We encourage practitioners to apply our methods responsibly, and to combine automated evaluation with careful human assessment when system outputs may have ethical or societal implications.

### 6.2 REPRODUCIBILITY STATEMENT

To ensure reproducibility, we design our framework to be lightweight and compatible with open-source MT systems and APIs. Our experiments are conducted on widely used, publicly available datasets and standard evaluation benchmarks. The ExpectoSample strategy is sampling-based and requires no fine-tuning or additional model training, lowering the barrier for replication. We hope that this can facilitate more research on these important topics in the academic community, as well as make our methods easier to replicate. We make all of our code and dataset available under an MIT license.

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

# 7 APPENDIX

## .1 USE OF LLMS

We use Claude-Sonnet-4[3] and GPT-5-Chat[4] to provide lexical and writing suggestions on the language part of this work; there is no direct usage with the output of LLMs in this paper without any modification.

# A MODEL USING

Table 5: Model Architecture and Size Overview

| Model Name | Size | Architecture |
|---|---|---|
| **NMT** | | |
| mbart (Lewis et al., 2020) | 610M | Dense |
| nllb-200-distilled (Costa-jussà et al., 2022) | 600M | Distill |
| nllb-200 (Costa-jussà et al., 2022) | 3.3B | Dense |
| nllb-moe (Costa-jussà et al., 2022) | 54.5B | MoE |
| **LLM (pre-trained only)** | | |
| Llama-2 (Touvron et al., 2023) | 7B | Dense |
| Qwen2.5 (Yang et al., 2024) | 7B | Dense |
| Llama-3.1 (Dubey et al., 2024) | 8B | Dense |
| Llama-2 (Touvron et al., 2023) | 70B | Dense |
| Llama-3.1 (Dubey et al., 2024) | 70B | Dense |
| Qwen2.5 (Yang et al., 2024) | 72B | Dense |
| **LLM (instruction-tuned)** | | |
| Llama-2 (Touvron et al., 2023) | 7B | Dense |
| Qwen2.5 (Dubey et al., 2024) | 7B | Dense |
| Llama-2 (Yang et al., 2024) | 70B | Dense |
| Qwen2.5 (Yang et al., 2024) | 72B | Dense |
| MiniCPM-MoE (Hu et al., 2024) | 8x2B | MoE |
| **LLM (reasoning)** | | |
| Qwen3-N Yang et al. (2025) | 8B | Dense |
| Qwen3-NT Yang et al. (2025) | 8B | Dense |
| DeepSeek-R1-Distill-Qwen-7B DeepSeek-AI et al. (2025) | 7B | Dense |
| DeepSeek-R1-Distill-Llama-8B DeepSeek-AI et al. (2025) | 8B | Dense |
| DeepSeek-R1-0628 DeepSeek-AI et al. (2025) | 671B | MoE |

As shown in Table 5, we systematically consider current SOTA MT systems encompassing NMT, LLM-based MT (pre-trained only, instruction-tuned, and reasoning) across different model size.

# B PROMPTS

## B.1 FOR INSTRUCTION-TUNED LLM

```
User:
Translate the following <source language> text to <target language>.
Only provide the translation, no explanations:

<source sentence>
```

## B.2 PROMPT ON PRE-TRAINED LLM

```
User:
Translate the following <source language> sentences to <target language>:

<source language>: 今天天气很好。
```

---

[3]https://www.anthropic.com/claude/sonnet
[4]https://chatgpt.com/

```
<target language>: The weather is beautiful today.

<source language>: 你好吗？
<target language>: How are you doing?

<source language>: 我期待着我们明天的会议。
<target language>: I'm looking forward to our meeting tomorrow.

<source language>: 技术的快速发展显著改变了我们的日常生活。
<target language>: The rapid development of technology has changed our daily lives
    significantly.

<source language>: 你能帮我解决这个问题吗？
<target language>: Could you please help me with this problem?

<source language>: <source sentence>
<target language>:
```

## C DEGREE OF NON-DETERMINISM OF ALL DATA

Table 6: Degree of Non-determinism Analysis with Percentage Values

| Model Name | Size | Degree of Non-determinism | | | | | |
| --- | --- | --- | --- | --- | --- | --- | --- |
| | | INNER | | COMET20DA | | COMET22KIWI | |
| | | mean | std | mean | std | mean | std |
| **NMT** | | | | | | | |
| MBART | 610M | 67.33 | 11.64 | 79.25 | 4.76 | 73.81 | 4.68 |
| NLLB-200 | 600M | 63.94 | 13.87 | 73.14 | 7.43 | 65.05 | 8.42 |
| NLLB-200 | 3.3B | 57.86 | 14.92 | 73.04 | 8.81 | 63.62 | 10.25 |
| NLLB-moe | 54.5B | 53.54 | 17.76 | 71.68 | 9.16 | 60.75 | 10.51 |
| **LLM (pre-trained only)** | | | | | | | |
| Llama-2 | 7B | 46.23 | 13.39 | 70.67 | 8 | 63.8 | 8.31 |
| Qwen2.5 | 7B | 56.89 | 19.64 | 77.45 | 7.6 | 75.09 | 5.85 |
| Llama-3.1 | 8B | 61.43 | 16.13 | 80.36 | 5.82 | 74.68 | 5.4 |
| Llama-2 | 70B | 52.24 | 13.65 | 70.73 | 8.61 | 67.98 | 7.03 |
| Llama-3.1 | 70B | 80.95 | 13.38 | 75.12 | 6.17 | 77.27 | 3.77 |
| Qwen2.5 | 72B | 61.22 | 16.47 | 76.37 | 11.43 | 78.11 | 3.56 |
| **LLM (instruction-tuned)** | | | | | | | |
| Llama-2 | 7B | 58.35 | 5.19 | 66.61 | 7.89 | 58.51 | 10 |
| Qwen2.5 | 7B | 85.65 | 10.00 | 85.15 | 2.06 | 79.01 | 2.1 |
| Llama-2 | 70B | 83.31 | 16.57 | 51.22 | 12.39 | 48.49 | 12.28 |
| Qwen2.5 | 72B | 90.35 | 7.78 | 86.85 | 1.34 | 80.59 | 1.09 |
| MiniCPM-MoE | 8x2B | 84.85 | 8.99 | 84.54 | 2.76 | 78.49 | 2.43 |
| **LLM (reasoning)** | | | | | | | |
| Qwen3(NT) | 8B | 90.66 | 8.05 | 86.24 | 1.2 | 80.68 | 0.99 |
| Qwen3 | 8B | 81.39 | 10.13 | 86.06 | 2.32 | 80.44 | 1.8 |
| DeepSeek-R1 | 7B | 63.51 | 12.36 | 80.27 | 5.53 | 73.77 | 6.31 |
| DeepSeek-R1 | 8B | 66.99 | 13.18 | 81.54 | 4.88 | 75.49 | 4.96 |
| DeepSeek-R1 | 671B | 70.56 | 11.84 | 84.86 | 3.02 | 80.33 | 2.42 |

