# OpenReview forum: "Understanding Before Evaluation: A Reliable Framework for Assessing Non-Deterministic Machine Translation Systems"
_ICLR.cc/2026/Conference — ICLR 2026 Conference Withdrawn Submission_

### Official Review · Reviewer_219F · 2025-10-28

**Soundness:** 2
**Presentation:** 1
**Contribution:** 2
**Rating:** 2
**Confidence:** 4

**Summary:**

The paper studies the inherent non-determinism in MT generation systems. Specifically, the non-determinism of these systems can affect the ranking of these MT systems which can pose risks of incorrect ranking when the systems can produce different translations due to non-determinism. The paper proposes lexical and semantic similarity metrics to measure diversity of translation generations.

**Strengths:**

The paper explores an important direction in machine translation system evaluation. The non-determinism problem for MT systems has not been widely explored. Thus this research direction is worthwhile.

**Weaknesses:**

1. It seems the metric calculated in equation 3, the inner overlap score is unnormalized. It should be normalized to get rid of the effect of vocabulary size and translation length.
2. The consistency of rankings should be measured with a consistency metric. However table 3 just reports the individual rankings with different sample sizes. Consistency of rankings for different sample sizes (including greedy) for each MT evaluation metric should be provided.
3. No method for selecting the best generation among the candidates have been provided. Using a MT-evaluator to select the best generation would be worth exploring.

**Questions:**

1. What do the last two columns of table 1 represent? Dataset? Semantic diversity?
2. In practice, how do you think we can select among the different generations of a MT system? Just having multiple generations without a system to choose the best one does not help. Thus from a practical perspective, greedy generation maybe the only viable approach and the ranking of greedy generation maybe the only one that is relevant. Here a pairwise MT translation evaluator may help as proposed in MT-ranker (https://arxiv.org/abs/2401.17099), to choose the best translation among the generations.
3. The paper explores an important research direction but the presentation quality is poor. As a first step to improving the paper the authors should expand their explanation of their tables.

---

### Official Review · Reviewer_jN3a · 2025-10-29

**Soundness:** 1
**Presentation:** 1
**Contribution:** 1
**Rating:** 2
**Confidence:** 4

**Summary:**

The paper proposes to add a step before the evaluation which takes the variance of the model into account and proposes ExpectoSample to mitigate the problem.
The sampling temperature, the easiest parameter to control the output variance, is set once in the whole paper and is ignored afterwards. Greedy decoding, the most straightforward solution to avoid variance is also ignored. The whole paper layout feels very sloppy.

**Strengths:**

Draws attention to the fact the sampling can have a big impact on the translation quality.

**Weaknesses:**

This paper is about variance in the system output, but then they set the temperature once and leave it like that. The chosen temperatures might alter the performance of different systems differently. It’s strange that the most important parameter for the output variance is completely ignored by this paper.

What are the actual results of the system? They are only reporting the position in the ranking compared to other systems. A table showing min, avg, max per system would have taken the same space, but contains much more information.

There is a very easy solution to the core problem in this paper: Greedy sampling. This should at least be part of the comparison.

**Questions:**

Line 104: The line starts with ‘=.’ which doesn’t seem to belong there.

Line 252: empty space between ‘ and s

Line 282: You got a subsection without text and only one subsubsection, I assume that is not what you want.

Line 323: No need to have the same footnote twice.

---

### Official Review · Reviewer_TTAF · 2025-11-01

**Soundness:** 1
**Presentation:** 2
**Contribution:** 1
**Rating:** 2
**Confidence:** 4

**Summary:**

This paper studies non-determinism and its effects in Machine Translation (MT) evaluation by first measuring diversity/variance in lexical outputs, and then measuring expected evaluation scores under multiple samples. The key point that this paper makes is that non-determinism can't be avoided and should hence be part of evaluations and system rankings. This is supported by evaluations on NMT/LLM models for en-zh translation. It claims providing a "efficient sampling strategy" powering a "lightweight" evaluation.

The main contribution is in my eyes the empirical quantification of effects of temperature sampling across models. In my opinion, i can serve as a motivation to prefer deterministic decoding algorithms for evaluation. The technical contributions of the paper, regarding the quantification of the variance are not substantial (averages, standard deviation, with standard diversity/similarity/evaluation metrics).

**Strengths:**

- The paper makes the point that non-determinism in decoding can affect system rankings - which is not unexpected but also not generally accounted for. This is a valuable point of reference for future works e.g. for developing better deterministic decoding strategies.
- The paper includes multiple MT systems in the evaluation, including classic MT systems and pretrained and finetuned LLMs.

**Weaknesses:**

- Framing: Non-determinism comes from the choice of decoding strategy, not from the model itself. If the authors had chosen temperature=0 (or close to 0 to be exact), the findings might have changed in that the variance and effect on rankings would have been much smaller, perhaps even negligible (compare e.g. https://arxiv.org/html/2402.06925v3).
- Unsupported claims: The paper claims to propose a "lighweight" method and an "efficient sampling strategy" - but the proposed method, repeated sampling with N samples, is neither lightweight (increasing evaluation costs by a factor of N), nor is it a new sampling technique, nor is its efficiency discussed or benchmarked in any way. The proposed solution consists of averages and expected evaluation scores - which is not at all novel or in any way made efficient.
- Narrow evaluation: The evaluation is conducted on a single language pair and domain, and for a single decoding strategy (temperature sampling) under one specific setting. It is clear that variance will differ for language pairs across models, and that the decoding strategy will affect the diversity and quality to expect (e.g. see https://arxiv.org/abs/2506.20544). For a wider picture, that the authors are trying to draw, it would be necessary to study the effect of these specific choices. The risk of narrow evaluations is exemplified on the concrete example that the authors provide, concluding that "for applications prioritizing semantic quality and stability, Qwen3-8B-NT (Yang et al., 2025) is the preferred choice" - from the evaluations under a single temperature on a single language pair on a single domain, which might not at all hold under a different decoding strategy or in a domain that the model is less familiar with.
- Clarity: I don't fully understand the effect of the two stages: how does the 'understanding stage" affect evaluations? How should authors, upon measuring non-determinism, act upon it? The paper is missing actionable suggestions here.

**Questions:**

- How would the conclusions change if the setup was slightly different, e.g. different language pair, decoding configuration, domain?
- What are users expect to do with the information about expected quality or average diversity? What are the implications and actionable steps that should be taken based on it?

---

### Official Review · Reviewer_kCA4 · 2025-11-01

**Soundness:** 2
**Presentation:** 2
**Contribution:** 2
**Rating:** 4
**Confidence:** 4

**Summary:**

The paper investigates evaluation of non-deterministic MT systems. The contributions are measuring non-determinism and automatic evaluation on samples instead on one single translation hypothesis which leads to more consistent rankings.

**Strengths:**

Making automatic metrics more reliable in the context of many possible output is an interesting topic.

**Weaknesses:**

The important concepts are not explained fully clear (see Questions for details)

Also, the results are not fully discussed and explained

There are no examples

**Questions:**

the problem with non-determinism for evaluation metrics is not fully clear

BLEU is mentioned, but no other metrics

what about metrics without references?

furthermore, there are no examples to illustrate

235: we adopt => we use

251: temperature' s => temperature's
or even better, effects of temperature

256-264: are those metrics used as the Sim function in Equation (4)?

Table 1: which similarity metric is used for the reported INNER score?
or all metrics in the paragraph 256-264 are used and somehow aggregated?

also, how were the two COMET variants used, directly?

furthermore, are the scores referring to the lexical non-determinism or semantic?

356: expectoSample is the average value of scores on different samples, right?

386: BLEU has different ranking than other metrics for all sample sizes: this is not mentioned?

388: the behaviour of TER is not discussed/explained?


related work should be placed after introduction, not at the end

---

### Note · Authors · 2025-12-03

I have read and agree with the venue's withdrawal policy on behalf of myself and my co-authors.